# Plasticity towards Rigidity: A Macrophage Conundrum in Pulmonary Fibrosis

**DOI:** 10.3390/ijms231911443

**Published:** 2022-09-28

**Authors:** Ezgi Sari, Chao He, Camilla Margaroli

**Affiliations:** 1Department of Medicine, Division of Pulmonary, Allergy & Critical Care Medicine, University of Alabama at Birmingham, Birmingham, AL 35294, USA; 2Department of Pathology, Division of Cellular and Molecular Pathology, University of Alabama at Birmingham, Birmingham, AL 35294, USA

**Keywords:** idiopathic pulmonary fibrosis, activated macrophages, fibrotic macrophages, EMT

## Abstract

Idiopathic pulmonary fibrosis (IPF) is a progressive, chronic, and ultimately fatal diffuse parenchymal lung disease. The molecular mechanisms of fibrosis in IPF patients are not fully understood and there is a lack of effective treatments. For decades, different types of drugs such as immunosuppressants and antioxidants have been tested, usually with unsuccessful results. Although two antifibrotic drugs (Nintedanib and Pirfenidone) are approved and used for the treatment of IPF, side effects are common, and they only slow down disease progression without improving patients’ survival. Macrophages are central to lung homeostasis, wound healing, and injury. Depending on the stimulus in the microenvironment, macrophages may contribute to fibrosis, but also, they may play a role in the amelioration of fibrosis. In this review, we explore the role of macrophages in IPF in relation to the fibrotic processes, epithelial–mesenchymal transition (EMT), and their crosstalk with resident and recruited cells and we emphasized the importance of macrophages in finding new treatments.

## 1. Introduction

Idiopathic pulmonary fibrosis (IPF) is a progressive, chronic, and ultimately fatal diffuse parenchymal lung disease. It is the most common interstitial lung disease (ILD) affecting mostly the subpleural parenchyma. IPF is characterized by irregular fibrotic patches, fibroblastic foci, and honeycomb changes [1,2,3]. The mortality rate in IPF approaches 50% within 3–5 years after the initial diagnosis [4,5], with survival rates lower than in some cancer patients [6]. In spite of critical scientific advances, the current metrics for diagnosis and treatment (published in 2022), are still not sensitive enough to identify patients with IPF in the early stage of the disease, which contributes to its poor prognosis [7].

While the molecular mechanisms of fibrosis in IPF patients are not fully understood, it is believed that repeated injuries to the alveolar epithelium may lead to dysregulated repair mechanisms and aberrant deposition of extracellular matrix [8,9]. Following the injury in the alveolar epithelium, recruited macrophages can contribute to the pathogenesis of pulmonary fibrosis via the secretion of a plethora of mediators such as cytokines, interleukins, transforming growth factor beta (TGF-β), connective tissue growth factor (CTGF), epidermal growth factor receptor (EGFR) ligands, and proteases [10,11,12]. Here, we review how molecular mechanisms of macrophage plasticity may orchestrate the onset of IPF and contribute to its progression (Figure 1), as well as current therapeutics and potential future macrophage-directed therapies.

## 2. Development of Pharmacological Treatment for IPF

Liebow et al. first described the histological feature of the usual interstitial pneumonia (UIP) pattern: coexisting fibrosis and interstitial/airway inflammation [13]. For decades, IPF has been considered as a chronic inflammatory condition in which repeated injuries lead to chronic scaring/fibrosis, where excessive oxidative stress due to chronic inflammation plays a key role in the pathogenesis. Hence, steroids have been the cornerstone for maintenance therapy as early data showed that high-dose steroids led to clinical improvement in a small population of patients (<30%) diagnosed with IPF [14]. Subsequently, Raghu et al. examined whether adding azathioprine, another potent immunosuppressive medication, to prednisone could benefit patients with IPF [15]. In this small (27 patients), double-blinded, randomized, placebo-controlled trial, prednisone and azathioprine combination therapy trended towards improved survival and reduced rate of decline in lung function measured by pulmonary function test (PFT). However, none of these parameters reached statistical significance. In addition to the escalation of immunosuppressive therapy, researchers have evaluated the efficacy of combining antioxidants with immunosuppressive medications. Behr et al. conducted an open-label trial with high-dose N-acetylcysteine (NAC) in 18 patients with or without immunosuppressive medications [16]. They found that 12 weeks of high-dose NAC treatment improved lung function index, a composite endpoint incorporating several PFT parameters. Moreover, patients who benefited most from NAC therapy were those on immunosuppressive medications. The follow-up double-blinded, randomized, placebo-controlled Idiopathic Pulmonary Fibrosis International Group Exploring N-Acetylcysteine I Annual (IFIGENIA) trial examined the addition of NAC or placebo to prednisone and azathioprine (the standard care arm) and found that adding NAC to prednisone/azathioprine slowed the decline of lung function. However, no survival benefits were observed [17]. The encouraging data from the IFIGENIA trial led to the NIH-sponsored double-blinded, randomized, placebo-controlled Prednisone, Azathioprine, and N-Acetylcysteine: A Study That Evaluates Response (PANTHER) trial [18]. In PANTHER, investigators evaluated the benefits of a combined triple therapy of prednisone, NAC, and azathioprine, compared with placebo. The trial was terminated earlier due to increasing mortality and rate of acute exacerbation in the experimental group compared with the placebo arm.

The failure of the PANTHER trial, together with two failed randomized Phase III trials evaluating interferon gamma-1β [19,20], marked a significant shift in developing therapies for IPF. Advances in basic and translational science into the molecular mechanism of IPF showed that IPF is a complex disease condition that begins with repeated epithelial injury and leads to recruitment of pro-fibrotic macrophages and activation of (myo)fibroblasts. Macrophages involved in the pathogenesis of IPF are no longer considered solely pro-inflammatory, but profibrotic and generate multiple growth factors. Consequently, there has been significant interest in developing anti-fibrotic, rather than anti-inflammatory, therapeutics targeting growth factor such as TGF-β. The subsequent double-blinded, randomized, placebo-controlled trials examining the two novel anti-fibrotics, namely pirfenidone and nintedanib, demonstrated a significantly reduced decline in lung function compared to placebo for both drugs [21,22,23]. FDA approved both medications for the treatment of IPF in 2014, and both received conditional recommendation from ATS/ERS practice guidelines in 2015 [24]. The exact target of pirfenidone remains unclear [25]. Initially developed as an anti-inflammatory medication, it was found to affect fibrosis development in various animal models. Nintedanib, a tyrosine kinase inhibitor, was initially designed for cancer treatment. It targets multiple growth factor signaling pathways such as fibroblast growth factor receptor-1 (FGF-1), vascular endothelial growth factor receptor-2 (VEGFR-2), and platelet-derived growth factor receptor (PDGF-R) α and β [26]. More recently, nintedanib has been approved to be used to treat systemic sclerosis-associated interstitial lung disease (SENSCIS) and progressive fibrosing interstitial lung disease (INBUILD) [27,28,29]. The effectiveness of anti-fibrotics in these non-IPF conditions highlights the shared profibrotic pathways in various ILDs and targeting profibrotic mediators can modulate ILD progression.

## 3. Macrophages in Fibrosis

Macrophages are central to lung homeostasis as well as in the orchestration of the immune response following an insult [30,31,32,33], which makes them an attractive target for future therapies. Therefore, here we will explore their role in IPF in relation to the fibrotic processes, epithelial–mesenchymal transition (EMT), and their crosstalk with resident and recruited cells.

Macrophages are an important part of host defense, and they are at the interface between innate and adaptive immunity [34]. Macrophages constitute one of the first lines of defense against external pathogens and insults [35], and are crucial for the phagocytic clearance of microorganisms, apoptotic cells, and cancer cells [36,37,38]. Over the decades it has been thought that the recruitment and activation of inflammatory cells contribute to pulmonary fibrosis. Indeed, recent studies have shown that the disturbed wound healing process and the activation of fibrotic responses can increase the activation and polarization of macrophages and lymphocytes [39]. It is also reported that IPF patients have increased levels of alveolar macrophages [32], making these cells a crucial player in fibrosis.

There are two distinct groups of macrophages in the lungs: interstitial macrophages (IMs) and alveolar macrophages (AMs) [40,41,42]. IMs are located within the lung parenchymal tissue, and they have regulatory functions including, tissue remodeling and maintenance of homeostasis. AMs are present in the alveoli, and they are the most abundant resident immune cells in lung homeostasis. They are involved in the phagocytosis of the external particles and maintain the surfactant catabolism in the alveoli [43,44,45,46,47,48]. Although the ontology of lung macrophages is still debated, it is thought that macrophages in the lung have three different developmental waves in the mouse lung tissue [49]. The first wave of macrophages is developed from yolk sac precursors, and they spread throughout the lung interstitium during embryonic development. In the second wave, the fetal liver-derived macrophages migrate to the alveoli and become alveolar macrophages. Third-wave macrophages are derived from bone marrow and reside in the lung interstitium during homeostasis [49]. According to their activation status, macrophages can be divided into two major subsets: classical activation (M1, pro-inflammatory/cytotoxic) and alternative activation (M2, anti-inflammatory/wound repair) [50,51]. Both populations can be further subdivided based on their function and gene expression: M2a works on phagocytosis, M2b is responsible for immunoregulation, and M2c plays a role in tissue modification and matrix deposition [50,52]. While these subsets are very well defined in the mouse, the dichotomy between the different macrophage phenotypes becomes less clear in the human lung. Nevertheless, their polarization states highlight the plastic potential of these cells.

Since macrophages are one of the main regulators of the immune response, their activation status might influence other cells’ behavior [53]. In IPF, it has been suggested that the microenvironment during injury may affect how monocytes differentiate into alveolar macrophages and promote tissue damage or fibrosis [54]. Indeed, several studies showed that monocytes can be drawn to the lung, where they can differentiate into Monocyte-derived AMs (Mo-AMs) and be alternatively activated towards the profibrotic or “M2” phenotype [55,56,57]. 

In IPF, CD163^+^ M2 macrophages are enriched in the fibrotic areas of the human lung [58]. Single cell analysis showed that tissue-resident alveolar macrophages, tissue-resident peribronchial and perivascular interstitial macrophages, and monocyte-derived alveolar macrophages are located in the fibroblastic foci. Further, single cell RNA sequencing revealed the presence of a Mo-AM population with pro-fibrotic gene signatures in an in vivo model of lung fibrosis. Presence of these Mo-AMs was demonstrated to be dependent upon signaling via the macrophage colony-stimulating factor receptor (M-CSFR), as the absence of M-CSFR depleted their presence in the fibrotic niches and ameliorated the pathology [59]. 

In the bleomycin-induced pulmonary fibrosis model, inflammatory macrophages increase immediately, reach the peak level on day 3, and slowly reduce until day 21. M1-like alveolar macrophages are the major population in the bronchoalveolar lavage (BAL) fluid at a steady state, however, after bleomycin exposure, M2-like alveolar macrophages gradually increased and reached the maximum level on day 14 which correlated with collagen deposition [31,60]. In an elegant study, Misharin and colleagues further highlighted the importance of Mo-AMs in the development of fibrosis. With a series of experiments using genetically engineered in vivo models, they were able to discern the role of Mo-AMs in a model of lung fibrosis. Briefly, Mo-AMs, but not AMs, contributed to the development of lung fibrosis in response to bleomycin and TGF-β via expression of pro-fibrotic genes, as necroptosis of Mo-AMs, attenuated bleomycin-induced lung fibrosis [54]. Further, they validated the expression of those genes in the human lung, showing the translational implication of the in vivo findings. Lastly, they showed that Mo-AMs persisted in the lung for a year after the initial fibrotic insult, highlighting a potential pathological mechanisms that could lead to aberrant responses to future insults.

In other models of fibrosis relevant to IPF’s risk factors, macrophages played a major role in the fibrotic processes after herpesvirus infection [61], and during radiation-induced lung fibrosis (RIF) [58]. Overall, in fibrosis macrophages play a pivotal role in the orchestration of the profibrotic response. Interactions between macrophages and other immune cells [62], as well between macrophages and parenchymal and mesenchymal lung cells, such as fibroblasts and epithelial cells, coupled with the degradation of the extracellular matrix (ECM) [63,64,65], contribute to fibrogenesis.

### 3.1. Cytokines, TGF-β and Wnt/β-Catenin Signaling

Macrophage polarization depends on the microenvironment. Toll-like receptor 4 (TLR4) signaling through Myd88-dependent IRAK-M expression has been shown to be increased in peripheral blood cells from idiopathic pulmonary fibrosis patients compared to controls and was linked to the alternative macrophage activation, profibrotic phenotype, and collagen production [66,67]. Further, it has also been reported that TLR signaling could trigger the pathology of pulmonary fibrosis by increasing the production and release of cytokines such as IL-6, TNF-α, and IL-1β [68]. In IPF, patients display higher levels of IL-6 in plasma and alternatively activated macrophages [31,69,70]. Upon TLR4 stimulation, alveolar macrophages start to release IL-6, IL-1β, and TNF-α, which trigger alveolar type 2 cells to secrete IL-1β and TNF-α helping recruit more leukocytes to the scar area [71], therefore fueling the inflammatory processes linked to fibrosis. While the ligand(s) responsible for TLR4 activation remain poorly defined in IPF, it will likely be a scenario where a combination of Pathogen-Associated Molecular Patterns (PAMPs) and Damage-Associated Molecular Patterns (DAMPs) will be required for the increased TLR4-mediated signaling [67,72,73,74,75,76].

Of the other cytokines potentially involved in IPF, IL-4, IL-10, and IL-13 represent another pathological loop that could fuel the fibrotic process. IL-10 expression has been linked to an increase in the T helper type 2 response, which modulates the production of the profibrotic cytokines IL-4 and IL-13 [77]. In the post-irradiation lung fibrosis model, IL-4 production by macrophages exacerbated the fibrotic process [78], while in the bleomycin model, blockade of IL-4 reduced fibrosis. In another model of IPF, IL-4 mediated the phosphorylation of STAT6, which regulated the M2 polarization via a redox-dependent mechanism, independent of Th2, and contributed to fibrosis [79,80]. Further, IL-13 contributes to fibrosis by changing the extracellular matrix (ECM) structure via the activation of ECM-degrading enzymes and promoting the fibrotic signaling pathways [81,82,83]. Although macrophages promote the wound healing processes by this signaling cascade [10], prolonged cytokine production deteriorates epithelial injury and promotes fibrosis. Long term inflammation might affect fibrosis like a major fibrotic process mediator TGF-β and modulating cytokines production from macrophages might be useful to ameliorate the pathology. 

TGF-β is a multifunctional regulatory protein acting on the SMAD family signaling [84,85]. It plays a role in a plethora of processes, including wound healing, inflammation, EMT, myofibroblast activation, collagen deposition, and apoptosis, and it is a major stimulator of fibrosis [33,86,87,88,89,90,91,92]. TGF-β is released by activated fibroblasts, platelets, and lymphocytes, including macrophages [90,93,94,95]. At homeostasis, it modulates macrophage differentiation in an autocrine fashion [33] via TGF-βR signaling, which is upregulated by PPAR-γ. However, in inflammatory conditions it can polarize macrophages towards an alternative activation state in an autocrine manner [95,96]. In human alveolar macrophages, stimulation with IL-4 and/or IL-10 increased the activation of STAT3, which led to the production of TGF-β [70]. Likewise, IL-13 increases TGF-β production by macrophages via IL-3Rα and AP-1-mediated activation of the TGF-β promoter [81]. TGF-β production is also regulated by the Wnt signaling pathway, one of the essential triggers of pulmonary fibrosis, and a major regulator of epithelial cell fate during development and injury [97,98]. In the bleomycin-mediated pulmonary fibrosis, deletion of the Wnt signaling pathway co-receptor LPR5, reduced TGF-β production AT2 cells and macrophages [99]. 

In the IL-4-treated M2 macrophages and epithelial cell co-culture system, TGF-β was increased in epithelial cells and EMT was triggered by TGF-β /Smad2 signaling pathway [100]. Interestingly, Murray et al. showed that depletion of M2 macrophages decreased collagen levels and attenuated fibrosis even in the presence of TGF-β [101], suggesting an important role for macrophage-derived IL-13, as highlighted by Borthwick and colleagues [102]. Indeed, in another study, the fibrosis regressed in IL-13^−/−^ mouse group even though TGF-β levels were same as in the wild type controls [93]. Overall, macrophage-derived TGF-β and IL-13 may contribute to chronic fibrosis in a two-hit model and affect the polarization of other macrophages, as well as key fibrotic mechanism such as EMT, myofibroblast differentiation, collagen deposition, and re-vascularization [31,103,104]. 

### 3.2. Role of Macrophages in Fibrogenesis, Myofibroblasts Differentiation and Epithelial–Mesenchymal Transition 

EMT is a tissue remodeling process in which epithelial cells lose their junctional structures (such as E-cadherin, and ZO-1) and gain distinct components of mesenchymal cell proteins and remodel the ECM (α-SMA, N-cadherin, and Vimentin) [105,106,107]. EMT contributes to fibrosis, invasion, and tumor metastasis [107,108]. In the late stage of injury, recruited macrophages by the extracellular microenvironment contribute to EMT [109,110,111]. Classical activation of macrophages and the release of its associated inflammatory factors increases EMT-related transcription factor levels, such as NF-κB, Snail, and Slug, and their nuclear localization in epithelial cells [110].

In co-culture experiments with classically or alternative-activated macrophages and epithelial or mesothelial cells, the levels of E-cadherin were downregulated, while α-SMA was upregulated. IL-8 increased significantly in the macrophage co-culture system and N-Cadherin and vimentin expression was elevated by JAK2/STAT3/Snail signaling pathway [112]. Moreover, epithelial cells underwent EMT, and concomitant further polarization to M2 was observed, suggesting two-way crosstalk between the epithelium and the macrophages that could worsen the pathology of IPF [64,100,113,114]. In cancer, CD68^+^ macrophages located around the tumor core contributed to the EMT process by triggering the downregulation of E-cadherin and the upregulation of vimentin, N-cadherin, and Snail in the epithelial and cancer cells via the release of TGF-β [112,115,116], a process potentially present in IPF as well. In M1 macrophages and mesothelial cells co-culture, the TRIF-dependent TLR4 signaling pathway was activated in the mesothelial cells to produce α-SMA and cells underwent the EMT process [114]. Overall, these studies show that inflammatory and pro-fibrotic cytokines released by activated macrophages in the microenvironment, induce the expression of EMT markers. Understanding the effect of activated macrophages on other cells in the fibrotic niche is an important area of research, as it could impact not only the development of IPF, but also the long-term inflammation-related progression of fibrosis. Further, focusing on the macrophage-related EMT process and modulating the cross-talk between macrophages and other cells in the niche might be a critical component of future therapeutics.

Activated macrophages promote fibrogenesis by releasing growth factors and triggering the proliferation and collagen synthesis of fibroblasts [117,118]. Single-cell RNA sequencing experiments in lung tissue from patients with fibrotic lungs showed co-expression of monocyte/macrophage and myofibroblast markers [119]. Dysregulated wound healing processes and aberrant cross-talks between macrophages and myofibroblasts may accelerate the progression of fibrosis. The crosstalk between these two cell types promotes myofibroblast activation resulting in fibrosis [10,120], while the accumulation of ECM components mediated by myofibroblasts increases the recruitment of macrophages [65]. This process is partly mediated by Cadherin-11 (CDH11), an adhesion protein. After adhesion via CDH11, macrophages can produce TGF-β and trigger myofibroblast differentiation in the fibrotic lung tissue in mice and humans [121]. In the macrophage-fibroblast co-culture with LPS-activated macrophages, fibroblasts secrete more matrix metalloprotease 7 (MMP7) and TNF-α; while during the co-culture with IL-4 activated macrophages fibroblasts express collagen, and TGF-β [118], thus promoting the pathological progression of the disease. Further, it has been reported that M2 macrophages influence myofibroblast differentiation and increase α-SMA levels by activation of the Wnt/β-catenin signaling pathway [31,122]. IL-4/IL-13 increases the phosphorylated STAT6 and JAK1 and up-regulates FIZZ1, which is a protein responsible for myofibroblast differentiation [123]. 

Although data are limited in IPF, it has been shown that bone marrow-derived macrophages can also increase the number of myofibroblasts via macrophage to myofibroblast transition (MMT), as reviewed in [124]. In the unilateral ureteral obstruction-induced pulmonary fibrosis model, 30% of myofibroblasts were CD68^+^ and α-SMA^+^, and alternatively activated macrophages were shown to directly transdifferentiate into collagen-producing α-SMA^+^ myofibroblasts via TGF-β/Smad3 signaling [113,119,125,126,127,128]. MTT also was also observed via STAT1, STAT3, and NF-ĸB signaling pathways in lung fibrosis [119], suggesting the cooperation of several signaling pathways. A potential mechanism of action for MMT has been elucidated in the bleomycin and TGF-β induced pulmonary fibrosis in vivo models, where the lung myofibroblasts induced epigenetic modifications in the tissue macrophages via histone lactylation, skewing the polarization of the alveolar macrophages into the pro-fibrotic phenotype [129]. Altogether, these studies indicate that macrophages can both stimulate fibroblast-myofibroblast differentiation, as well as potentially contribute to the pathology via MTT. 

### 3.3. Role of Macrophages on ECM

The ECM microenvironment controls homeostasis and regulates cell adhesion, migration, cell cycle, metabolism, and cell differentiation [130]. ECM degradation occurs during fibrosis by matrix metalloproteases (MMPs) and other enzymes [131,132]. Dysregulation of the deposition and degradation of ECM promotes the formation of the fibrotic niches [133], and targeting the pathways regulating ECM degradation and collagen degeneration in fibrosis could be a promising treatment [134].

The effect of macrophages on ECM is a double-edged sword. Macrophage-derived MMPs exert antagonistic roles in fibrosis with regard to the regulation of the ECM and the progression of IPF, reviewed in [133]. In the bleomycin-induced pulmonary fibrosis model, TGF-β1 was overexpressed by macrophages and mesenchymal cells and ECM deposition occurred [94]. Overexpressed TGF-β1 can increase the ECM deposition by reducing the ECM degradation through reduction the antifibrotic MMP1 and induction of TIMP1 (MMP inhibitor) [135]. On the other hand, a fibrotic active-MMP7 is abundantly increased in IPF patients, and it is localized on the surface of alveolar macrophages [136]. Further, MMP28 contributes to pulmonary fibrosis and M2 polarization in the lung [137,138]. In the late stage of injury, macrophages overexpress MMP-9, which contributes to EMT via osteopontin cleavage [109]. MMP12, a macrophage-secreted elastase, is highly induced in the lung in response to injury and regulates fibrosis by controlling the expression of MMP-2 and MMP-13 in IL-13Rα-dependent manner. The loss of MMP-12 increased the level of MMP-2 and MMP-13 and ameliorated fibrosis [138]. Further, macrophage-released MMP-14 and MMP-9 can also clear the fibrotic area in the tissue by breakdown the extracellular matrix and digesting the collagen [104,134], while milk fat globule EGF 8 (MFGE8)-positive macrophages can phagocytose the collagen and attenuate fibrosis [139]. Depending on the status of the signals in the ECM, macrophages may contribute to fibrosis by secreting MMPs that increase TGF-β release in a fibrotic environment, or they may play a role in the clearance of fibrosis by taking part in collagen degradation. Clarifying the underlying signals and molecular mechanisms of such dichotomy may prove to be critical for the development of effective therapies.

### 3.4. Macrophage-Derived Extracellular Vesicles 

Exosomes are small (30–150 nm) membrane vesicles and are released into the extracellular environment [140]. Exosomes and extracellular vesicles (EVs) carry RNA, including microRNA (miRNA), transfer RNA (tRNA), ribosomal RNA (rRNA), messenger RNA (mRNA), and additional structural and non-coding RNAs, as well as effector proteins. They regulate cell–cell signaling [141,142]. EV-carried miRNA can contribute to fibrosis [143,144,145] as highlighted by Let-7 miRNA. Let-7 targets IGF1, which is overexpressed in alveolar macrophages and can stimulate collagen production in IPF [146,147]. Further, in macrophages, Let-7 can change the M1-M2 polarization in the fibrotic lung [148]. IL-4 and IL-13 upregulate miR-142-5p and downregulate miR-130a-3p in macrophages so that they can sustain the profibrogenic effect [149]. As Kishore and Petrek discussed in their review, macrophages can join the EMT and ECM deposition by regulating fibrotic signaling pathways via miRNA upregulation/downregulation [150]. The polarization of M1 alveolar macrophages by TGF-β1 reduced the level of miR-124 and over-expression of miR-124 suppressed M1 alveolar macrophage polarization [151]. Macrophage-Derived miR-21-5p-containing exosomes increased the α-SMA, TGF-β1, and collagen accumulation in the resident cells in Smad7 dependent manner and contributed to the fibrosis [152].

EVs also showcase proteolytic enzymes on their lipid capsules, which can cleave surface proteins and degrade the ECM [153,154]. MMP14+ EVs degrade collagen, fibronectin, and vitronectin in the ECM and activate MMP-2, which is another collagen-degrading protease. Further, in the absence of EV-delivered MMP14, the pulmonary fibrosis process deteriorated [155,156,157]. Macrophage-delivered exosomes also have a disintegrin and metalloproteinase, ADAM10, and ADAM15 on their surface and they cleave and activate TNFα, EGF, and collagen IV [157,158,159]. Further, ADAM10 can transactivate EGFR via G protein-coupled receptors, increasing the likelihood of the development of pathological settings in IPF [160]. Even though the ADAM family works on collagen degradation as MMPs [157,161], TGF-β also promotes ADAM10 expression and myofibroblast activation occurs via ADAM10-mediated sEphrin-B2 generation. Further, pharmacological inhibition of ADAM10 prevents lung fibrosis in mice [162]. Likewise, smoke increased ADAM15 expression in alveolar macrophages and in airway α-SMA-positive cells, while both ADAM10 and ADAM15 shed E-cadherin and changed the cell membrane structure [163,164,165]. Overall, the current studies show that macrophage-derived EVs are an important factor in establishing cell-to-cell communication and reorganizing the extracellular matrix via collagen degradation, cleavage of membrane proteins, stimulation of fibrotic signaling pathways, and activation of macrophages. Progression of fibrosis can be modulated by the different cargos in the EVs, therefore understanding their role in IPF may prove critical. 

### 3.5. Cell Senescence and Genetic Factors

IPF is an age-related disease. Indeed, cellular senescence, telomere shortening, epigenetic modifications, and mitochondrial dysfunction can contribute to the aging mechanisms in IPF [166,167,168,169,170]. During aging, phagocytotic cells start to lose their function, basal innate immune signaling is increased and the injury-related responses and cellular defense are decreased. As the macrophage phenotype shifts with ageing, coupled with enhanced oxidative metabolism, loss of migration capabilities, their basal release of inflammatory factors may impact their pathological imprint on the tissue [166,167,168,169,170]. Indeed, with aging, the alveolar microenvironment displays a constant inflammatory state, characterized by increased release of IFN-γ, TNF-α, IL-12, IL-1β, IL-6, and IL-10 by aged macrophages [171]. Moreover, senescence of the alveolar epithelial cell type 2 (ATII) increases IGF-1 expression, and polarization of M2 macrophages through IL-13 signaling [172]. On the other hand, plasminogen activator inhibitor 1-mediated TGF-β1-induced ATII cell senescence activates alveolar macrophages and contributes to lung fibrogenesis by inducing p16 [173]. Old macrophages can release more IL-10, a profibrotic cytokine, than young ones [174]. Macrophage migration inhibitory factor (MIF) is related to telomere shortening and ECM deposition, and it has been reported that MIF levels are higher in the fibrotic niches in the lung tissue and that MIF is increased in the BAL of IPF patients compared to controls [175,176]. Overall, IPF is an age-progressive disease and with the progressive ageing of the world population its incidence may increase in the years to come, making efforts to understand its pathological mechanisms even more crucial.

## 4. Macrophages as a Target

Targeting lung macrophages as a therapeutic might be useful for the treatment of fibrosis. Some studies show that blockade of the recruitment of the monocyte-derived macrophages in the fibrotic niche might attenuate the fibrotic processes [54,177,178]. A biomarker of lung transplantation/death CHI3L1 is produced by airway epithelial cells, alveolar type II epithelial cells, and macrophages in the lung, and the level of CHI3L1 is elevated in IPF patients [55,179]. It induces the M2 differentiation and their recruitment to the scar. Further, CHI3L1 recruits CD206^+^ mouse lung macrophages, which can stimulate fibroblast proliferation [55], making it an interesting therapeutic candidate and warranting more investigations.

Targeting macrophages at the cytokine level could be achieved via several pathways in addition to those mentioned in the previous sections. For example, serum amyloid P (SAP) targets collagen deposition via stimulating the release of IP10/CXCL10 by M2 macrophages and ameliorates bleomycin-induced pulmonary fibrosis [180,181]. SAP can control the peripheral blood monocyte differentiation and their activation states by binding to Fcγ receptors [180]. Another route could be via the targeting of the CCR4 signaling axis. Clinical and in vivo studies showed that CCR4 and its ligand CCL17 are regulators for fibrosis in the lung [182,183]. CCL17 led to M1 activation and oxidative injury by inducing NOS2. Further, in the absence of CCR4 in macrophages, the oxidative lung injury by bleomycin was decreased [183]. Likewise, the CCL2/CCR2 pathway are increased in the fibrotic niche and CCR2 deficiency decreases the fibrotic area through macrophage-derived MMP-2 and MMP-9 production [180,184]. Lastly, the Colony-Stimulating Factor Receptor-1 (CSF1R) is one of the signals that controls M1/M2 polarization towards M2. In the radiation-mediated pulmonary fibrosis model, investigators showed that activated IMs were able to induce myofibroblastic activation and ECM production via CSF1R activation, as treatment with the CSF1R mAb reversed the phenotype [58]. 

Targeting redox signaling and mitochondrial biogenesis in the lung macrophages might be another therapeutic approach. NOX4 is essential for macrophage polarization for fibrotic repair in the lung by inducing the production of profibrotic molecules for collagen deposition [178,185]. Further, Th2-independent, redox-dependent M2 polarization by regulation of STAT6 provides a potential therapeutic target for attenuating the progression of pulmonary fibrosis. STAT6 mediates Cu, Zn-SOD–induced M2 polarization in a redox-dependent manner by regulating the Jumonji domain containing (Jmjd) 3. Both STAT6 and Jmjd3 deficiency reduced the M2 polarization. Further, Leflunomide treatment reduced mitochondrial reactive oxygen species production and inhibited Jmjd3 expression and M2 polarization. Taken together, these observations provide evidence that the redox regulation of STAT6 and Jmjd3 is a unique regulatory mechanism for profibrotic M2 polarization [56,79].

Lung transplantation is the only current effective treatment for IPF. Although two antifibrotic drugs (Nintedanib and Pirfenidone) are approved and used for the treatment of IPF, they have side effects and they do not improve the survival rate [1,186]. Therefore, understanding the role of lung macrophage functions and using them as a potential therapeutic targets could prove critical. Pirfenidone inhibits pro-inflammatory cytokines (TNF-α, IL-1, IL-6, etc.) and promotes the production of anti-inflammatory cytokines (IL-10). However, this drug potentially targets fibroblasts and it can only delay the disease progression [187]. Macrophages express MMP2 and MMP9 that degrade collagen, and pirfenidone blocks the alternative activation of macrophages and modulates MMP activity [22,188]. Nintedanib blocks the polarization of both M1 and M2 human macrophages and markers that contribute to lung fibrosis (IL-1β, IL-8, IL-10, and CXCL13) by altering colony-stimulating factor 1 (CSF1) receptor (CSF1R) activation and its downstream PI3K/Akt signaling pathway [189]. 

Recently, the rescue and reprogramming of macrophages is becoming a novel therapeutic approach for the treatment of lung diseases, as highlighted by the treatment with the HIF-1α stabilizer, which promoted macrophage adaptation to the hypoxic microenvironment via their glycolytic metabolism, which resulted in the protection of the lungs from inflammation-induced injury [190]. To this end, several drugs have been shown to have an effect on the modulation of macrophage activity. Although not all of them have been used in IPF, their effect on macrophages could be relevant to pulmonary fibrosis. Indeed, some currently used treatments could interfere with the pathological fibrotic process, as detailed in Table 1, while other ones may exacerbate aberrant pathways (Table 2).

## 5. Conclusions

In conclusion, macrophages promote collagen and ECM degradation, therefore reducing fibrosis, while on the other hand they can stimulate the fibrotic signaling pathways by inducing EMT, MTT, and Wnt signaling. IPF is a difficult disease to treat and there has been extensive research on this subject. In clinical studies, either there are insignificant results or the drugs may do more harm than good. Further, the pivotal role of macrophages on this disease is undeniable. However, more studies are needed to answer critical questions (Figure 2) leading to a better understanding of the mechanisms and inform more effective treatment strategies.

## Figures and Tables

**Figure 1 ijms-23-11443-f001:**
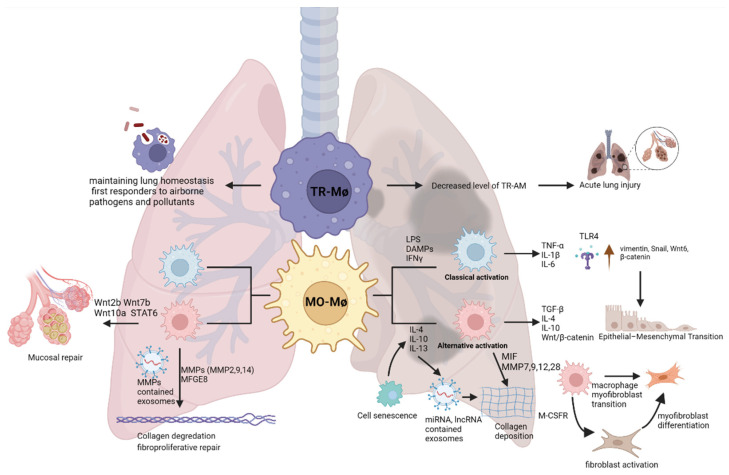
Schematic representation for hypothetical the role of alveolar macrophages in pulmonary fibrosis. The left side of the figure represents the how macrophages can clean the fibrotic foci and keep homeostasis, the right side shows macrophage-induced fibrosis.

**Figure 2 ijms-23-11443-f002:**
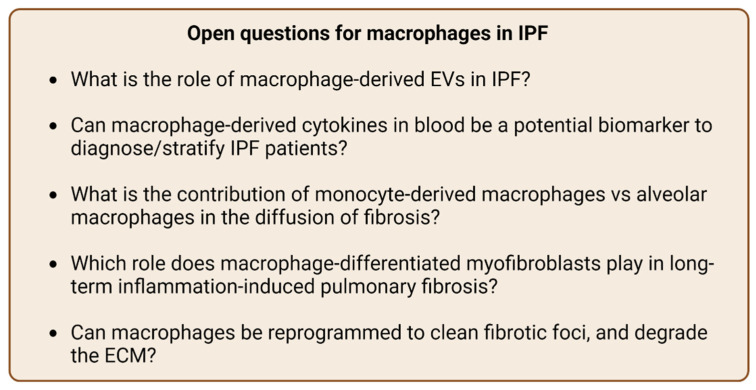
Open questions related to macrophage-related idiopathic pulmonary fibrosis.

**Table 1 ijms-23-11443-t001:** Therapeutics that could ameliorated the fibrotic process via macrophage activation.

Treatment	Mechanisms of Action	Effect on Macrophages
Atorvastatin	Lipid-decreasing statin	Reduces mediator production of AM (IL-1β, IL-6, and TNF-A-α) and macrophage recruitment [191,192]
Artemisinin	Antiviral, antimalarial, and anti-inflammatory	Inhibits macrophage chemotaxis and cytokine production [193]
Artesunate	Antimalarial	Attenuates proinflammatory effects of monocytes/macrophages [194]
Chloroquine	Immunosuppressive and Anti-parasite	Reduces TNF-A-α, IL-1-β and IL-6 [195,196]
Corticosteroid	Anti-inflammatory	Reduces macrophage CD64, CD80 and CD86 expression, controls the phenotype of alveolar macrophages [197].
Cyclophosphamide	Chemotherapy and Immunosuppressive	Activates and enhances macrophage phagocytosis [198]
Dasatinib	Chemotherapy	Elevates production of IL-10 while suppressing the production of IL-6, IL-12p40 and TNF-α in response to TLR stimulation [199]
HDAC6 Nexturastat A	HDAC inhibitor	Reduces pro-tumorigenic M2 macrophages [200].
Hydroxychloroquine	Immunosuppressive and Anti-parasite	Promotes apoptosis of macrophages and inhibits activation of macrophages, especially M2 macrophages [201]
Infliximab	Immunosuppressive	Induces apoptosis of Ly6C^+^ macrophages, decreases migration of monocytes into the ankles, and reduces CCL2 [202]
Intravenous Immunoglobulin (IVIG)	Therapy treatment for patients with antibody deficiencies	Inhibits the activation of monocytes and macrophages, inhibition of macrophage responses to IFN-γ [203]
PD-1/PD-L1 signaling blocker	Checkpoint inhibitor anticancer drug	Decreases TNF-A-α, IL-6, IFN-γ and ROS from alveolar macrophages [204]
Suberoylanilide hydroxamic acid (SAHA), Vorinostat	Chemotherapy	Reduces TNF-α, IL-1-β, IL-12, and IFN-γ [205]
Tocilizumab	Immunosuppressive	Anti-IL-6 receptor, modifies macrophage activation [206]
Zanubrutinib	Kinase inhibitor	Inhibits M1 macrophage polarization and promotes M2 macrophage polarization [207]

**Table 2 ijms-23-11443-t002:** Therapeutics that could exacerbate the fibrotic process via macrophage activation.

The Name of Drug	The Type of Drug	Effect on Macrophages
Amiodarone	Antiarrhythmic	Induces alveolar macrophages to secrete more TNF-α and superoxide anions [208,209].
Bleomycin	Chemotherapy	Recruits pro-fibrotic M2 cells and induces myofibroblast differentiation [31,66]
Cyclophosphamide	Chemotherapy and Immunosuppressive	Decreases spontaneous proliferation and reduces the ability to proliferate upon stimulation with GM-CSF [210]
Docetaxel	Chemotherapy	Induces M2 cells recruitment [211]
Methotrexate	Chemotherapy and Immunosuppressive drug	Macrophage recruitment [212]
Procainamide	Antiarrhythmic	Induces macrophage recruitment [213]

## Data Availability

Not applicable.

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
