# Peer review of "Plasticity towards Rigidity: A Macrophage Conundrum in Pulmonary Fibrosis"

_ijms, 2022, doi:10.3390/ijms231911443_

Round 1
Reviewer 1 Report
The aim of the authors was to present a review entitled “Plasticity towards rigidity: a macrophage conundrum in pulmonary fibrosis”. Writing a review it is a big work which involves bringing new and important data related to the subject-matter addressed.
The review focus on the role of macrophages in Idiopathic Pulmonary fibrosis process and their crosstalk with resident and recruited cells. Current therapy and open questions for a future therapeutic strategy targeting macrophages are also discussed.
The subject of the manuscript is interesting, overall well written, comprehensive presented and it faces an interesting issue with possible biomedical applications.
In conclusion the manuscript fits with the scope of the journal and the authors have done a good work finally. Based on my comments the manuscript can be published in the present form.
Author Response
We thank the reviewer for these encouraging comments and thorough review.
Reviewer 2 Report
In general, this was an informative, well-written manuscript. Some suggested changes/clarifications are as follows:
L13: change "immunosuppressive" to "immunosuppressants"
L30: change "inspite" to "in spite"
L41-42 change "could" to "may"; delete "how do they"
L49: change "steroid has" to "steroids have"
L50: change "steroid" to "steroids"
L64: What does IFIGENA stand for? Describe.
L69: What does PANTHER stand for? Dscribe.
L85: delete "medicines"
L98: insert "the" after "orchestration of"
L116: What do you mean by "natural"? Do you actually mean "resident"?
L151: add "fluid" after "BAL".
L153: change "got" to "reached"
L160: change "gene" to "genes"
L186: "this patient's population" What patient and what population? Explain.
L250-252: This sentence doesn't make sense, please correct.
L254: change "as well" to "also"
L380-381: change "part" to "side"
L383: change "macrophage" to "macrophages"; delete "agent"
L424: change "target" to "targets"; correct spelling of "coupld"; change "inhibit" to "inhibits"
L425: change "promote" to "promotes"
L428: change "modulate" to "modulates"
L486-506: delete all of the excess "the"
L509: change "macrophages" to "macrophage"
l510: "they contain vacuolated cytoplasm" doesn't make sense here - please correct.
Author Response
We thank the reviewer for these encouraging comments and their help to improve our manuscript. Below see attached the list of revisions made to the manuscript.
